# CAPM: Fast and Robust Verification on Maxpool-based CNN via Dual Network

## Abstract

This study uses CAPM (Convex Adversarial Polytope for Maxpool-based CNN) to improve the verified bound for general purpose maxpool-based convolutional neural networks (CNNs) under bounded norm adversarial perturbations. The maxpool function is decomposed as a series of ReLU functions to extend the convex relaxation technique to maxpool functions, by which the verified bound can be efficiently computed through a dual network. The experimental results demonstrate that this technique allows the state-of-the-art verification precision for maxpool-based CNNs and involves a much lower computational cost than current verification methods, such as DeepZ, DeepPoly and PRIMA. This method is also applicable to large-scale CNNs, which previous studies show to be often computationally prohibitively expensive. Under certain circumstances, CAPM is 40-times, 20-times or twice as fast and give a significantly higher verification bound (CAPM 98% vs. PRIMA 76%/DeepPoly 73%/DeepZ 8%) as compared to PRIMA/DeepPoly/DeepZ. (cf. Fig. 3 and Fig. 4). Furthermore, we additionally present the time complexity of our algorithm as $O(W^2 N K)$, where $W$ is the maximum width of the neural network, $N$ is the number of neurons, and $K$ is the size of the maxpool layer's kernel.

## 1 Introduction

In the past few years, convolution neural networks have reached unprecedented performance in various tasks such as face recognition (Hu et al., 2015; Mehdipour Ghazi & Kemal Ekenel, 2016) and self-driving cars (Rao & Frtunikj, 2018; Maqueda et al., 2018), to name a few. However, these networks are vulnerable to malicious modification of the pixels in input images, known as adversarial examples, such as FGSM (Goodfellow et al., 2015), PGD (Madry et al., 2018), One Pixel Attack (Su et al., 2019), Deepfool (Moosavi-Dezfooli et al., 2016), EAD (Chen et al., 2018), GAP (Poursaeed et al., 2018), MaF(Chaturvedi & Garain, 2020) and many others (Wong et al., 2019).

In view of the threat posed by adversarial examples, how to protect neural networks from being tricked by adversarial examples has become an emerging research topic. Previous studies of defense against adversarial examples are categorized as *Removal of adversarial perturbation* (Akhtar et al., 2018; Xie et al., 2019; Jia et al., 2019; Samangouei et al., 2018) and *Adversarial training* (Shafahi et al., 2019; Han et al., 2020; Tramer et al., 2020). Both defense mechanisms may protect the network from certain adversarial examples but there is only empirical evidence that they do so. The robustness of the network is not guaranteed. It is impossible to train or evaluate all possible adversarial examples so these methods are vulnerable to other adversarial examples that are not in the data sets that are used.

The need for guaranteed robustness assessments has led to the development of verification mechanisms for a neural network. These verify specific properties pertaining to neural networks, such as robustness against norm-bounded perturbation (Dvijotham et al., 2018; Singh et al., 2018), robustness against adversarial frequency or severity (Katz et al., 2017) and robustness against rotations (Singh et al., 2019a).

During the early development of neural network verification, satisfiability modulo theories (SMT) solver (Katz et al., 2017) and semidefinite programming (SDP) methods (Raghunathan et al., 2018) were used. A SMT solver yields tight verification bounds but is not scalable to contemporary networks with sophisticated

architecture. The SDP method requires less time but is limited to linear architectures. Recent studies have developed verification tools for more realistic scenarios, such as a fully connected neural network (FCNN) with an activation function and a convolution neural network (CNN). As indicated by Salman et al. (2020), the main methods for neural network verification can be categorized as either primal view or dual view.

The primal view method involves *Abstract interpretation* and *Interval bound propagation*. There are classic frameworks for abstract interpretation (e.g., AI2 (Gehr et al., 2018), DeepZ (Singh et al., 2018), DeepPoly (Singh et al., 2019a)). As a step further, RefineZono (Singh et al., 2019b) and RefinePoly (Singh et al., 2019b) use mixed integer linear programming (MILP) to improve the verification bounds for DeepZ and DeepPoly, respectively. However, the computation time that is required for verification is significantly increased. Another bounding technique for the primal view method is interval bound propagation, which uses interval arithmetic to obtain the bound for each individual neuron in each layer. Representative works include IBP (Gowal et al., 2018) and CROWN-IBP (Zhang et al., 2019). Dual-view methods (Wong & Kolter, 2018; Dvijotham et al., 2018; Wong et al., 2018; Bunel et al., 2020; Xu et al., 2020; Wang et al., 2021) formulate the verification problem as an optimization problem, so according to Lagrangian duality (Boyd et al., 2004), each dual feasible solution yields a lower bound to the primal problem and verification bounds are derived by solving the dual problem. Moreover, noteworthy among these methods is the state-of-the-art approach $\alpha,\beta$-CROWN (Wang et al., 2021), which is grounded in the LiRPA framework (Xu et al., 2020).

## 1.1 Verification of a CNN with maxpooling

The maxpool function is an integral part in most real-world neural network architectures, especially CNNs (e.g., LeNet (LeCun et al., 1998), AlexNet (Krizhevsky et al., 2012) and VGG (Simonyan & Zisserman, 2015)), which are widely used for image classification. However, past works have the following shortage in the verification of networks involving maxpool functions:

- *Not applicable:* IBP (Gowal et al., 2018) and others (Wong & Kolter, 2018; Wong et al., 2018) (Bunel et al., 2020; De Palma et al., 2021) verify a CNN but there is no theory to verify maxpool-based networks. Gowal et al. (2018) analyzed several monotonic activation functions (e.g., ReLU, tanh, sigmoid) in IBP but they did not consider non-monotonic functions (e.g., maxpool). Wong & Kolter (2018) discussed the verification of a ReLU-based FCNN and a later study (Wong et al., 2018) uses this for the verification of residual networks. However, besides referring to the work of Dvijotham et al. (2018), the study by Wong et al. (2018) does not address much about handling maxpool functions. Bunel et al. (2020); De Palma et al. (2021) analyzed networks with nonlinear activation functions, such as ReLU and sigmoid, but there is no analysis of the maxpool function.

- *Has theory but lack of implementation evidence:* These studies analyze the maxpool function but experiments only verifiy ReLU-based CNNs. Examples include AI2 (Gehr et al., 2018), DeepZ (Singh et al., 2018), DeepPoly (Singh et al., 2019a), RefineZono (Singh et al., 2019b), RefinePoly (Singh et al., 2019b) and LiRPA (Xu et al., 2020). Dvijotham et al. (2018) analyzed a large variety of activation functions, such as ReLU, tanh, sigmoid and maxpool, but they only demonstrated the experiment result on a small network consisting of one linear layer, followed by sigmoid and tanh. CROWN-IBP (Zhang et al., 2019) used the verification method of IBP (Gowal et al., 2018) and analyzed non-monotonic functions, including maxpool, but experiments only verify results for ReLU-based CNNs. In spite of providing functions corresponding to maxpool, LIRPA is currently unable to function properly on maxpool-based CNNs. The definitions of DenseNet and ResNeXt used in their experiments can be found in their GitHub repository, and it's noteworthy that these definitions do not include a maxpool layer.

- *Imprecise:* These studies give an imprecise verification bound for maxpool-based CNN. DeepZ (Singh et al., 2018), DeepPoly (Singh et al., 2019a), and PRIMA (Müller et al., 2022) implement the verification of maxpool-based CNN but experimental results are nevertheless lacking. For comparison purposes, we implement these methods on 6 maxpool-based CNN benchmarks modified from (Mirman et al., 2018) (cf. Supplementary Material A.4). Our experiment indicates that DeepZ, DeepPoly, and PRIMA are imprecise in our benchmarks. For a norm-bounded perturbation $\epsilon = 0.0024$, the

verified robustness for a convSmall CIFAR10 structure decreases to 1%, 25%, and 26%, respectively (cf. Fig. 3). Due to the implementation of the BaB (branch and bound) algorithm in $\alpha,\beta$-CROWN (Wang et al., 2021) on maxpool-based CNNs, it results in excessive GPU memory requirements (more than 13GB) or prolonged execution times (more than 5 minutes per example). Consequently, we consider examples that trigger the BaB algorithm as not verified.

- *Computational costly:* These studies involve a significant computational cost to verify each input image in the maxpool-based CNN benchmarks (cf. Sec. 3.1). Our experiment shows that for $\epsilon = 0.0006$ on convBig CIFAR10 (cf. Fig. 4), PRIMA (Müller et al., 2022) requires 6.5 days and DeepPoly (Singh et al., 2019a) requires 3 days to verify 100 images. (See Sec. 3.1 for hardware spec)

Yuan et al. (2019) showed that $l_\infty$ norm is one of the most commonly used perturbation measurements (e.g.,DeepFool (Moosavi-Dezfooli et al., 2016), CW (Carlini & Wagner, 2017), Universal adversarial perturbations (Moosavi-Dezfooli et al., 2017), and MI-FGSM (Dong et al., 2018)). Therefore, this study verifies a maxpool-based CNN with $l_\infty$ norm-bounded perturbation. The contributions of this study are:

- CAPM (Convex Adversarial Polytope for Maxpool-based CNN) is used to improve the verified bound for a maxpool-based CNN, assuming $l_\infty$ norm-bounded input perturbations. The maxpool function is decomposed into multiple ReLU functions (see Sec. 2.2), as such convex relaxation trick by Wong & Kolter (2018) can be applied. While we are not the first to consider the relationship between maxpool functions and ReLU functions in our paper, our experiments demonstrate that the algorithm we propose is currently the most effective among all executable methods.

- A dual network for general purpose CNNs is derived with maxpool, padding and striding operations to obtain the verified bound efficiently (cf. equation 48).

- The results (cf. Sec. 3.2) show that CAPM gives a verification that is as robust as the state-of-the-art methods (PRIMA), but there is significantly less runtime cost for MNIST and CIFAR 10.

- The experimental results show the limitations of a Monte-Carlo simulation for a large-scale neural network verification problem, which demonstrates the necessity for a provable robustness verification scheme.

- Among algorithms applicable to maxpool-based CNNs, we are currently the only ones providing a clearly defined time complexity for our algorithm. The time complexity of the CAPM algorithm is $O(W^2NK)$.

The main limitation of our algorithm is that it requires adherence to the assumptions defined in Supplementary Material. A.1.1. We address future work based on this in the conclusions section.

The reminder of this paper is organized as follows: Sec. 2 defines the verification problem and introduce the key idea of CAPM with a toy example. Sec. 3 compares the experimental result for CAPM against with DeepZ, DeepPoly, PRIMA, LiRPA and $\alpha,\beta$-CROWN in terms of the verified robustness and average runtime metrics for various adversary budgets. The experiment result also indicates that the accuracy lower bound predicted by each verification method becomes looser as norm-bounded perturbation increases. Conclusions and future work are summarized in Sec. 4. Furthermore, we gives a bound analysis for intermediate layers to explain the reason for this phenomenon in Supplementary Material A.3.

## 2 Method

This section illustrates the verification of a CNN classifier under the $l_\infty$ norm-bounded perturbations. Sec. 2.1 defines the verification problem and Sec. 2.2 gives an overview of CAPM. More details for solving the verification problem for a maxpool-based CNN are in Supplementary Material A.1 and Supplementary Material A.2.

Supplementary Material A.1 specifies the CNN architecture for this study; Supplementary Material A.2 describes the formulation of the the verification problem in terms of Lagranian dual problems, and simplifies the dual constraints to the form of a leaky ReLU dual network.

## 2.1 Definition of the verification problem

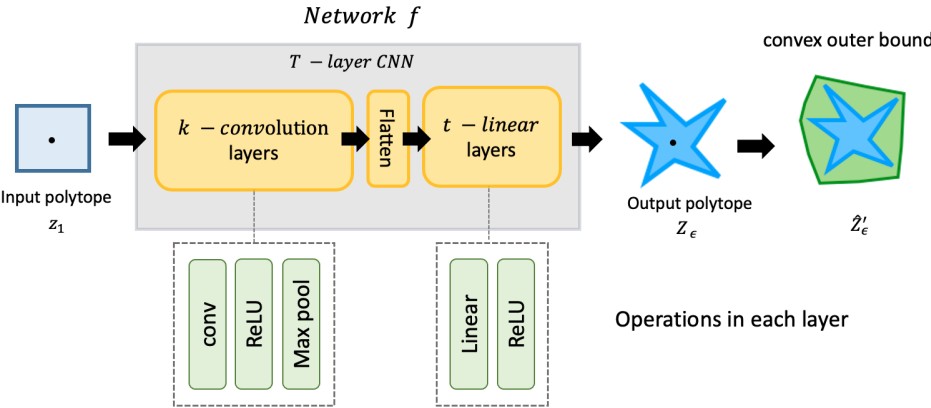

Figure 1: Adversarial polytope passes through network $f$

This study evaluates the robustness of a CNN to arbitrary adversarial examples within a bounded norm budget, to see whether or not the prediction result of the CNN will change under such adversarial perturbations. The verification problem is reformulated through convex relaxation as a convex optimization problem and the duality theorem states that any dual feasible solution can serve as a lower bound for the original verification problem.

A neural network $f$ consisting of $k$ convolution layers with ReLU activation and maxpool functions, followed by flattening and $t$ fully connected layers with ReLU activation is shown in Fig 1. If a clean image $x$ is added with perturbation $\Delta$, to which this study impose a $l_\infty$-norm constraint $\|\Delta\|_\infty \leq \epsilon$, the perturbed input $z_1$ resides in an input adversarial polytope that is described as:

$$\mathbf{z_1} \leq \mathbf{x} + \epsilon \quad \text{and} \quad \mathbf{z_1} \geq \mathbf{x} - \epsilon. \tag{1}$$

The perturbed input $z_1$ is taken as the input of the network $f$. Though at first glimpse the input adversarial polytope (cf. equation 1) is a hyper-cube which is convex, and that the ReLU function is itself a convex function, the intermediate adversarial polytope after passing the input adversarial polytope through the convolution and ReLU activation is in general not a convex set, as the set $\{(\xi, ReLU(\xi)) : \xi \in \mathbb{R}\}$ is not a convex subset of $\mathbb{R}^2$. The verification problem deviates from a convex optimization framework due to the non-linearity of ReLU and maxpool equalities. The presence of these non-linear equalities introduces complexity, rendering the optimization problem non-convex. Similarly, the application of a nonlinear function leads to a non-convex output polytope, even if the input polytope maintains convexity. In other words, substituting nonlinear equality constraints with linear inequality constraints results in the creation of a convex outer bound. To make the feasible solution a convex set, a convex outer bound (Wong & Kolter, 2018) is constructed when passing through each of the ReLU and maxpool functions. The output polytope $Z_\epsilon$, which is the collection of all possible results computed by $f$ at the output layer with a perturbed input $z_1$, is contained within a convex polytope $\hat{Z}'_\epsilon$. Both $Z_\epsilon$ and $\hat{Z}'_\epsilon$ are subsets of $\mathbb{R}^K$ for a $K$-class classification task.

For an image $x$ that is labeled with ground truth $y^* \in \{1, ..., K\}$ to which an adversary attempts to mislead network $f$ into falsely predicting a target label $y^{targ} \in \{1, ..., K\}$ rather than $y^*$, a necessary condition is that

the adversary must find a perturbed input $z_1$ satisfying equation 1 so that $\mathbf{e}_{y^*}^T f(z_1) \leq \mathbf{e}_{y^{targ}}^T f(z_1)$, where $\mathbf{e}_{y^*}$ and $\mathbf{e}_{y^{targ}}$ are one-hot encoded vectors of $y^*$ and $y^{targ}$, respectively. Therefore, as $f(z_1) \in Z_\epsilon \subset \hat{Z}'_\epsilon$, if the minimum for the optimization problem

$$\min_{\hat{\mathbf{y}} \in \hat{\mathbf{Z}}'_\epsilon} \quad (\mathbf{e_{y*}} - \mathbf{e_{y^{targ}}})^{\mathbf{T}} \hat{\mathbf{y}} \tag{2}$$

is positive for every target class $y^{targ} \in \{1, ..., K\} \setminus \{y^*\}$, then network $f$ cannot be fooled by an adversarial example that differs from image $x$ by a perturbation with at most $\epsilon$ under $l_\infty$-norm. This method can guarantee zero false negatives, so the system flags every image that is prone to attack by an adversarial example, but it may falsely flag some images resilient to perturbations.

### 2.2 CAPM overview

This section describes a toy example to illustrate the use of CAPM to solve the optimization problem in equation 2 using a Maxpool-based network. The dual problem is formulated using Lagrangian relaxation (Boyd et al., 2004) and convex relaxation (Wong & Kolter, 2018), so any dual feasible solution corresponds to a lower bound to the original problem in equation 2. Convex relaxation loosens the verification bound but Wong & Kolter (2018) showed that this lower bound can be calculated using a backpropagation-like dynamic programming process in a so called dual network. As the determination of upper and lower bounds for preceding layers constitutes a sub-problem within the dual network framework for subsequent layers, employing a dynamic programming algorithm becomes a viable approach to address this verification problem. This study extends the method of Wong & Kolter (2018) to a maxpool-based CNN and demonstrates that maxpool-based CNNs can also be verified efficiently and precisely using a dual network.

#### 2.2.1 Toy example

CAPM verifies the robustness of a simple maxpool-based network under the $l_\infty$ norm constraint in equation 1. The verification problem for this toy example is formulated as an optimization problem in equation 3. If the lower bound of equation 3 is positive for all possible target classes, then this network is not misled by any input perturbation, $l_\infty$-norm that is less than $\epsilon$. If not, then this network may not be safe for this input perturbation.

$$\begin{aligned}
\min_{\hat{\mathbf{z}}'_3} \quad & (\mathbf{e_{y*}} - \mathbf{e_{y^{targ}}})^{\mathbf{T}} \hat{\mathbf{z}}_3 \equiv \mathbf{d^T} \hat{\mathbf{z}}_3 \\
\text{s.t.} \quad & \mathbf{z}_1 \leq \mathbf{x} + \epsilon \\
& \mathbf{z}_1 \geq \mathbf{x} - \epsilon \\
& \hat{\mathbf{z}}_2 = \mathbf{W}_1 \mathbf{z}_1 + \mathbf{b}_1 \\
& \mathbf{z}_2^R = \max(\hat{\mathbf{z}}_2, \ \mathbf{0}) \\
& \mathbf{z}_2 = \max(z_{2,0}^R, \ z_{2,1}^R, \ z_{2,2}^R, \ z_{2,3}^R) \\
& \hat{\mathbf{z}}_3 = \mathbf{W}_2 \mathbf{z}_2 + \mathbf{b}_2
\end{aligned} \tag{3}$$

In equation 3, the perturbed input $\mathbf{z}_1$ is the input to a simple maxpool-based network. The feature map $\hat{\mathbf{z}}_2$ is obtained by inputting $\mathbf{z}_1$ into the linear operation in the fully-connected layer. ReLU and maxpool are then used to compute the intermediate results $\mathbf{z}_2^R$ and $\mathbf{z}_2$. The output $\hat{\mathbf{z}}_3$ is calculated using the linear operation. This is a non-convex optimization problem because of the non-affine activation functions ReLU and maxpool so Wong & Kolter (2018)'s method of convex relaxation (see Supplementary Material A.1.2 for more details) is applied to the ReLU function over the input interval, which approximates the ReLU function using the linear outer bounds (cf. Fig. 9). In terms of the maxpool function (see A.1.1 and A.1.2 for more details), $z_2 = \max(z_{2,0}^R, \ z_{2,1}^R, \ z_{2,2}^R, \ z_{2,3}^R)$ is decomposed into several one-by-one comparisons using dummy variables

$$z_{2,j+1}^M = \max(z_{2,j}^R, \ z_{2,j}^M) = z_{2,j}^M + \max(z_{2,j}^R - z_{2,j}^M, \ 0), \ j \in [\![0, 3]\!]. \tag{4}$$

As such, $z_{2,4}^M = \max(z_{2,0}^M, z_{2,0}^R, z_{2,1}^R, z_{2,2}^R, z_{2,3}^R)$, and $z_2 = z_{2,4}^M$ if one chooses $z_{2,0}^M$ no larger than the maximum of $z_{2,0}^R, ..., z_{2,3}^R$. In this example, $z_{2,0}^M = 0$ because all elements in $z_2^R$ are the output of ReLU so they are

non-negative, equation 4 is then split into several terms:

$$\bar{z}_{2,j} = z_{2,j}^R - z_{2,j}^M \tag{5a}$$

$$z'_{2,j} = \max(\bar{z}_{2,j}, 0) \tag{5b}$$

$$z_{2,j+1}^M = z'_{2,j} + z_{2,j}^M \tag{5c}$$

After decomposition using equation 5, the second term is also a ReLU function so convex relaxation is used, assuming knowledge of the upper-lower bounds of $\bar{z}_{2,j}$. The ReLU and maxpool functions in equation 3 are then replaced by convex outer bounds linear inequalitiy constraints to form a convex optimization problem. The dual problem can then be written in the form of a dual network (see Supplementary Material A.2.2 for detailed derivation):

$$\max J_\epsilon(\Theta) \quad \text{s.t.} \quad F(\mathbf{d}, \alpha^R, \alpha^M) \tag{6}$$

which has the form of a leaky-ReLU network. If the dual optimal of the convex relaxation problem is positive, then the system verifies the network as robust and if not, the system does not exclude the possibility that the network can be fooled by some perturbation with $l_\infty$-norm of at most $\epsilon$. The variables $\alpha^R$ and $\alpha^M$ are considered to be additional free variables for the dual network, the choice of which affects the precision of the verification bound that is calculated by the dual network. A strategy that is similar to CAP (Wong & Kolter, 2018) is used to determine $\alpha^R$ and $\alpha^M$.

### 2.2.2 Determining the upper-lower bound

The node-wise upper-lower bounds are required for the convex relaxation of ReLU functions. To determine the upper-lower bounds, namely $\hat{l}_{2,j} \leq \hat{z}_{2,j} \leq \hat{u}_{2,j}$, for the input nodes of the ReLU function, a verification problem is formulated that corresponds to the network up to the linear layer before the first ReLU. The node-wise bounds for $\hat{z}_{2,j}$ are determined by evaluating the resulting (smaller) dual network with one-hot input vector $\boldsymbol{e}_j$ (instead of $\mathbf{d}$) (Wong & Kolter, 2018). In terms of the element-wise lower and upper bounds, namely $\bar{l}_{2,j} \leq \bar{z}_{2,j} \leq \bar{u}_{2,j}$, that pertain to the maxpool functions, each maxpool function is decomposed into multiple ReLU activations (cf. equation 4). Thus computing these bounds layer-by-layer as in (Wong & Kolter, 2018) would be very costly. The values for $\bar{l}_{2,j}$ and $\bar{u}_{2,j}$ can be calculated more efficiently as follows: For (cf. equation 5a)

$$\bar{z}_{2,j} = z_{2,j}^R - z_{2,j}^M,$$

if the element-wise lower and upper bounds for $z_{2,j}^R$ and $z_{2,j}^M$, namely

$$l_{2,j}^R \leq z_{2,j}^R \leq u_{2,j}^R \quad \text{and} \quad l_{2,j}^M \leq z_{2,j}^M \leq u_{2,j}^M$$

are known, then the element-wise lower and upper bounds $\bar{l}_{2,j}$ and $\bar{u}_{2,j}$ for $\bar{z}$ are derived as

$$\bar{l}_{2,j} = l_{2,j}^R - u_{2,j}^M \quad \text{and} \quad \bar{u}_{2,j} = u_{2,j}^R - l_{2,j}^M.$$

The elementwise bounds $l_{2,j}^R$ and $u_{2,j}^R$ on the pre-maxpool activations can be computed in a way similar to how the elementwise bounds for pre-ReLU activations are computed in (Wong & Kolter, 2018). To compute $l_{2,j+1}^M$ and $u_{2,j+1}^M$, recall that

$$z_{2,j+1}^M = \max\left\{ z_{2,j'}^R : j' \in [\![0,j]\!] \right\}.$$

Thus, one can take

$$l_{2,j+1}^M = \max\left\{ l_{2,j'}^R : j' \in [\![0,j]\!] \right\}, \quad \text{and} \quad u_{2,j+1}^M = \max\left\{ u_{2,j'}^R : j' \in [\![0,j]\!] \right\}.$$

Since our algorithm design is primarily an extension of the method proposed by Wong & Kolter (2018), we can deduce the time complexity of our algorithm to be $O(W^2 N K)$ by analyzing that the time complexity of Wong & Kolter (2018)'s algorithm is $O(W^2 N)$.

# 3 Experiment

This section determines the verified robustness and the average verification time for CAPM, DeepZ, DeepPoly, PRIMA (Müller et al., 2022) and $\alpha,\beta$-CROWN for a $l_\infty$ norm-bounded perturbation of various budgets and for various attack schemes, such as FGSM and PGD. Sec. 3.1 details the experimental network architecture and the input dataset. Sec. 3.2 compares the results for this study with those of previous studies (DeepZ, DeepPoly, PRIMA and $\alpha,\beta$-CROWN) in terms of the verified robustness and the average verification time, and Supplementary Material A.5 illustrates how we reproduced the state-of-the-art methods so that we can have a fair comparison with them. Experiments in Supplementary Materials A.3.2 also demonstrate that the neural network verification problem cannot be simply evaluated using a Monte-Carlo simulation. All experiments were conducted on a 2.6 GHz 14 core Intel(R) Xeon(R) CPU E5-2690 v4 with a 512 GB main memory.

## 3.1 Experiment setting

### 3.1.1 Benchmarks

Robustness is calculated against adversarial examples on several networks that are trained using different methods:

- **Dataset:** Models are trained using the MNIST and CIFAR10 datasets. Images are normalized using the default setting for DeepPoly (Singh et al., 2019a). For MNIST, the mean and standard deviation is 0.5 and 0.5, respectively. For CIFAR 10, the mean and standard deviation of the RGB channels is $(0.485, 0.456, 0.406)$ and $(0.229, 0.224, 0.225)$, respectively.

- **Architecture of neural networks:** There are no empirical verificaion results on maxpool-based CNNs so maxpool layers are added to the common benchmark networks, convSmall, convMed and convBig in (Mirman et al., 2018). The parameter for striding and padding is adjusted to achieve a similar number of parameters to previous studies. Information about these 6 networks is shown in Table 1. The detail structures are shown in Supplementary Material A.4. Moreover, although the authors of $\alpha,\beta$-CROWN didn't conduct experiments in maxpool-based CNNs before and they did make some additional assumptions on their maxpool layers, we would like to compare with them in maxpool-based CNNs. Hence, we also create the network benchmark $conv_S$, $conv_M$ and $conv_L$ for their settings.

- **Training methods:** We compared the verification results of CNNs trained either normally (without adversarial training) or with adversarial examples such as Fast-Adversarial (Wong et al., 2020) and PGD (Madry et al., 2018).

- **Performance metrics:** The performance of neural network verification is often evaluated through the following metrics (Singh et al., 2019a):

  - *Verified robustness:* This is expressed as the number of images verified to be resilient to adversary example attack, divided by the total number of accurate images. This ratio represents the analysis precision of a verifier when a neural network is applied to a test image dataset that is subject to attack by an adversarial example.
  - *Average verified time:* This is the total time that is required by the verification algorithm to verify images, divided by the total number of images.

Table 1: Network parameters

| Dataset | Model | # Hidden layers | # Parameters |
|---------|-------|-----------------|--------------|
| MNIST | convSmall | 3 | 89606 |
| | convMed | 3 | 160070 |
| | convBig | 7 | 893418 |
| CIFAR10 | convSmall | 3 | 125318 |
| | convMed | 3 | 208198 |
| | convBig | 7 | 2466858 |
| MNIST | $conv_S$ | 3 | 9538 |
| | $conv_M$ | 4 | 19162 |
| | $conv_L$ | 4 | 568426 |

### 3.1.2 Robustness evaluation

For each test dataset, the settings for DeepPoly (Singh et al., 2019a) are used and the top 100 clean images are used as the evaluation test dataset. Adversarial examples are generated by adding to clean images with $l_\infty$ norm-bounded perturbation for various budgets $\epsilon$ and various attack schemes, such as FGSM and PGD. The generated adversarial examples are then applied to the neural network to compare the accuracy of lower bounds that are evaluated using various verification methods. A better verification method must give a tighter (higher) accuracy for the lower bound and never exceeds that of existing attack schemes. The implementation details for DeepPoly, PRIMA and $\alpha$,$\beta$-CROWN are described as follows:

- **DeepZono and DeepPoly**: We follow the implementation as suggested by the default command in their GitHub (Eth-Sri).

- **PRIMA**: We use exactly the same configuration mentioned in (Müller et al., 2022) for convSmall and convBig. Since PRIMA didn't report any results on convMed, and that convMed has the same number of layers as convSmall, we use the same configuration that PRIMA applied to convSmall on convMed as well.

- **$\alpha$,$\beta$-CROWN**: We used the default parameters and set the parameter conv mode to matrix in order to enable the operation of the BaB algorithm. The examples that would trigger the BaB algorithm were considered not verified. This adjustment was made because the implementation of the BaB algorithm on maxpool-based CNNs results in excessive GPU memory requirements (more than 13GB) and extended execution times (more than 5 minutes per example).

### 3.2 Experimental results

The results for CAPM are compared with those of previous studies (DeepZ, DeepPoly, PRIMA) in terms of the verified robustness and average runtime metrics for various adversary budgets $\epsilon$ for six different networks, as shown in Fig. 3 and Fig. 4. The classification accuracy [1] for a real-world PGD attack (orange solid line with triangle marker) is also determined. A verification method must demonstrate verified robustness that is no greater than the accuracy of real-world attack schemes. Fig. 3 and Fig. 4 illustrate that CAPM achieves better verification robustness than all other schemes for all combination of the CIFAR10 dataset and has a much lower computational cost. The computational cost of CAPM is independent of $\epsilon$, because a different adversary budget corresponds to the same dual network architecture so computational costs are similar. However, the verification time that is required by PRIMA increases as $\epsilon$ increases, possibly because as $\epsilon$ increases, the intermediate adversarial polytope becomes more complicated and must be described by a more complex MILP optimization problem. This demonstrates the promising potential of CAPM towards verification of large scale CNNs and colour images. Due to the excessive GPU memory requirements of $\alpha$,$\beta$-CROWN, we did not include $\alpha$,$\beta$-CROWN in this set of experiments.

---

[1]Here we follow the settings in (Singh et al., 2019a; Müller et al., 2022) and only test on images which were correctly classified before any perturbation is added.

The results for the MNIST dataset are shown in Table 2 and Table 3. PRIMA and CAPM both yield a higher (tighter) verification robustness than DeepPoly or DeepZ (cf. Fig. 2). CAPM also has a comparable verification robustness to PRIMA, but average runtime is significantly reduced. For larger networks such as convBig, only CAPM achieves effective verified robustness in a reasonable runtime.

For $\alpha,\beta$-CROWN, the BaB algorithm they employ is not suitable for maxpool CNNs, leading to excessive GPU memory requirements (more than 13GB) and prolonged execution times (more than 5 minutes per example). Therefore, in this experiment, we considered examples that would trigger the BaB algorithm as not verified. Consequently, the performance of $\alpha,\beta$-CROWN is not satisfactory in our study.

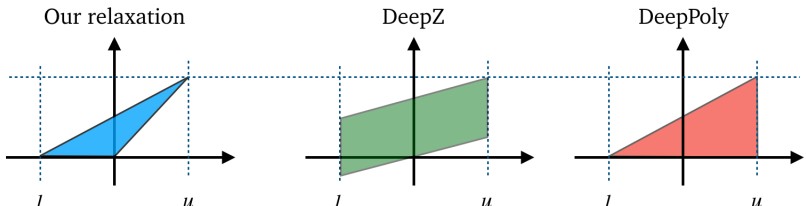

Figure 2: The difference in convex relaxation for the method of this study, DeepZ, and DeepPoly.

CAPM has the following advantages, compared to other state of the arts verification methods (DeepZ, DeepPoly, and PRIMA):

- CAPM achieves comparable or better verification robustness with significantly less runtime cost.

- Unlike other verification methods for which runtime increases with the adversary budget $\epsilon$, CAPM has a constant runtime that is independent of $\epsilon$. For larger networks such as convBig, only CAPM demonstrates a feasible runtime and good verified robustness.

- CAPM is especially suitable for larger-scale maxpool-based CNNs that are designed for color images. Using the CIFAR10 dataset, CAPM achieves a significantly tighter verified robustness at a much lower computational cost. And due to computational costs, we did not continue our experiments on larger and more general networks. Nevertheless, from Tables 2 and 3 in our experiments, it can be observed that our CAPM not only exhibits a lower growth rate in computation time as the neural network scales up but also maintains nearly equal accuracy under significantly lower computation time compared to PRIMA.

## 4 Conclusion

This study extends Wong & Kolter (2018)'s work to general purpose CNNs with maxpool, padding, and striding operations. The key idea for handing the maxpool function is to decompose it into multiple ReLU functions, while special care is taken to speed-up the computation of element-wise bounds required for the convex relaxation of intermediate ReLUs in maxpool layer. General purpose CNNs are expressed using a dual network, which allows efficient computation of verified bounds for CNNs.

The experimental results show that CAPM outperforms previous methods (DeepZ, DeepPoly, and PRIMA) in terms of verified robustness and computational cost for most adversary budget settings, and especially for large-scale CNNs for color images. For an adversary budget $\epsilon = 0.0024$, the verified robustness for DeepZ, DeepPoly and PRIMA for convSmall CIFAR10 decreases by to 1%, 25%, and 26%, respectively, but CAPM has a verified robustness of 87.5%; For an adversary budget $\epsilon = 0.0006$ for convBig CIFAR10, CAPM is 40-times and 20-times faster than PRIMA and DeepPoly, respectively, and gives a significantly higher verified robustness (see Fig. 3 and Fig. 4).

The proposed method gives comparable or better verification with significantly less runtime cost. Unlike many verification methods, for which runtime increases with the adversary budget $\epsilon$, CAPM has a constant

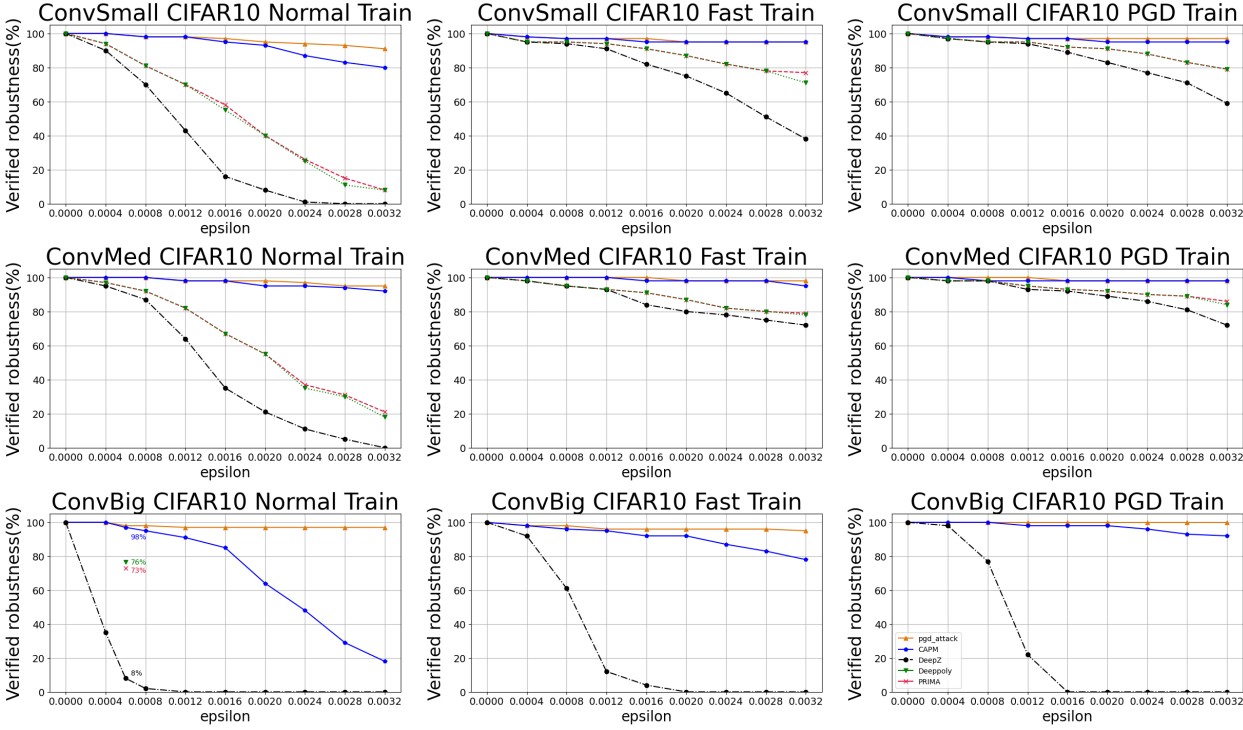

Figure 3: Verified robustness for $\epsilon$ perturbations under $l_\infty$-norm by CAPM (black solid line with pentagon marker), DeepPoly (green dotted line with triangle marker), DeepZ (black dashdot line with circle marker), and PRIMA (red dashed line with x marker) for convSmall CIFAR10, convMed CIFAR10, and convBig CIFAR10. The orange solid line with triangle markers is the classification accuracy for a PGD attack. PRIMA and DeepPoly are both prohibitively computationally costly for convBig CIFAR10, so we only show the result for DeepZ.

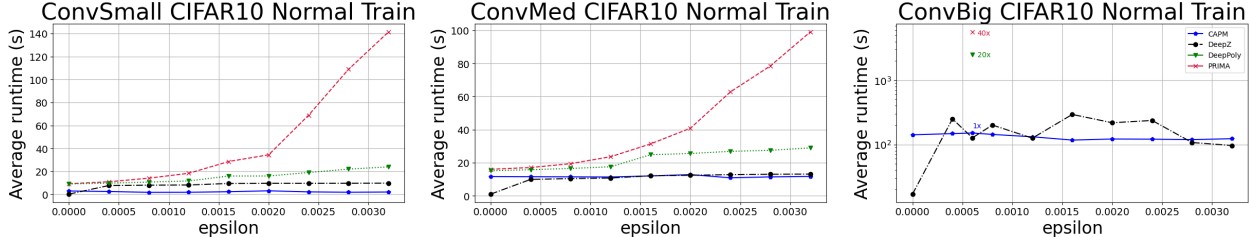

Figure 4: Runtime for $\epsilon$ perturbations under $l_\infty$-norm for CAPM (black solid line with pentagon marker), DeepPoly (green dotted line with triangle marker), DeepZ (black dashdot line with circle marker) and PRIMA (red dashed line with x marker) for convSmall CIFAR10, convMed CIFAR10 and convBig CIFAR10.

Table 2: Comparison with previous studies using the MNIST dataset

| Model | Training | Accuracy | $\epsilon$ | Our Ver | DeepZ Ver | DeepPoly Ver | PRIMA Ver |
|---|---|---|---|---|---|---|---|
| convSmall | Normal | 100 | 0.03 | 65 | 0 | 40 | **79** |
| convSmall | Fast | 100 | 0.03 | 85 | 2 | 72 | **92** |
| convSmall | PGD | 100 | 0.03 | **97** | 10 | 91 | **97** |
| convSmall | PGD | 100 | 0.04 | 84 | 0 | 53 | **85** |
| convMed | Normal | 100 | 0.01 | 93 | 89 | 95 | **96** |
| convMed | Fast | 100 | 0.03 | 88 | 49 | 88 | **96** |
| convMed | PGD | 100 | 0.03 | 96 | 82 | 96 | **97** |
| convMed | PGD | 100 | 0.04 | 89 | 15 | 87 | **93** |
| convBig | Normal | 100 | 0.01 | 73 | 0 | 86 | **86** |

| Model | Training | Accuracy | $\epsilon$ | Our Time | DeepZ Time | DeepPoly Time | PRIMA Time |
|---|---|---|---|---|---|---|---|
| convSmall | Normal | 100 | 0.03 | **1.14** | 3.76 | 9.47 | 229.8 |
| convSmall | Fast | 100 | 0.03 | **1.17** | 4.55 | 8.95 | 71.28 |
| convSmall | PGD | 100 | 0.03 | **1.23** | 3.68 | 6.82 | 25.35 |
| convSmall | PGD | 100 | 0.04 | **1.95** | 3.90 | 8.19 | 149.7 |
| convMed | Normal | 100 | 0.01 | **4.67** | 5.94 | 10.74 | 15.39 |
| convMed | Fast | 100 | 0.03 | **4.24** | 5.60 | 10.11 | 35.16 |
| convMed | PGD | 100 | 0.03 | **4.72** | 4.73 | 9.20 | 26.98 |
| convMed | PGD | 100 | 0.04 | **4.56** | 5.88 | 10.93 | 58.77 |
| convBig | Normal | 100 | 0.01 | 338 | **120.1** | 1493 | 5560 |

Table 3: Comparison with $\alpha, \beta$-CROWN using the MNIST dataset

| Model | Training | Accuracy | $\epsilon$ | Our Ver | $\alpha$-$\beta$-crown Ver |
|---|---|---|---|---|---|
| $conv_S$ | Normal | 98 | 0.03 | 66 | 0 |
| $conv_S$ | Fast | 99 | 0.03 | 94 | 33 |
| $conv_S$ | PGD | 99 | 0.03 | 95 | 37 |
| $conv_S$ | PGD | 99 | 0.04 | 93 | 11 |
| $conv_M$ | Normal | 99 | 0.01 | 97 | 13 |
| $conv_M$ | Fast | 98 | 0.03 | 95 | 13 |
| $conv_M$ | PGD | 100 | 0.03 | 96 | 7 |
| $conv_M$ | PGD | 100 | 0.04 | 90 | 2 |
| $conv_L$ | Normal | 99 | 0.01 | 95 | 0 |

runtime, regardless of the adversary budget, so it can be used for larger-scale CNNs which are usually computationally prohibitive for other verification methods. The proposed verification method is suited for use with large scale CNNs, which are an important element of machine learning services.

This study does provide a more precise and efficient verification for maxpool-based CNNs but the verified network is limited to a specific architecture that is defined in Supplementary Material A.1.1. Future study will involve the design of a verification framework that is applicable to neural networks with a more flexible architecture.

Additionally, the method of simplifying certain layers into multiple ReLU layers may likely be limited to maxpool layers only. Therefore, our preliminary future direction will focus on achieving greater flexibility in maxpool-based CNNs. Subsequently, we will continue exploring more flexible neural network architectures, such as residual connections.

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
