# OpenReview forum: "CAPM: Fast and Robust Verification on Maxpool-based CNN via Dual Network"
_TMLR — Rejected by TMLR_

### Review · Reviewer_Zine · 2024-07-22

**Summary Of Contributions:**

This paper presents a novel method, termed CAPM, for enhancing the verification bounds of maxpool-based CNNs under bounded norm adversarial perturbations. By decomposing the maxpool function into a series of ReLU functions, CAPM extends convex relaxation techniques to maxpool functions, allowing efficient computation of verification bounds through a dual network. Experimental results demonstrate that CAPM achieves SOTA verification precision with significantly lower computational costs compared to existing methods.

**Audience:**

Yes

**Broader Impact Concerns:**

None.

**Claims And Evidence:**

No

**Requested Changes:**

It is recommended that the authors further refine the paper, expand the experimental scope, and provide more detailed analysis and discussion to enhance the paper's persuasiveness and practicality.

**Strengths And Weaknesses:**

[Strengths]

The method proposed in this paper not only achieves accuracy comparable to SOTA methods but also significantly reduces algorithm runtime.


[Weaknesses]
1. The writing style of the paper is not clear enough, and some expressions are inaccurate, which may affect readers' understanding of the proposed method. Even in the toy example in Section 2.2.1, every symbol should have a clear definition, specifying whether it is a vector, matrix, or tensor, and what its dimensions are. Especially, each symbol should be clearly defined and detailed when it first appears. For example, $f \in \mathbb{R}^d$ represents a d-dimensional feature vector.

2. The authors should provide more detailed descriptions of the model architectures used in the experiments, such as convSmall, convMed, and convBig, including network depth, parameter count, etc. Otherwise, readers will not know how large convBig is compared to classic network architectures like VGG, ResNet, and DenseNet. This is crucial because the authors claim that "This method is applicable to large-scale CNNs". If the network architectures used in this paper are smaller than existing classic networks, the authors need to validate the method's effectiveness on larger network structures.

3. Figure 3 shows that the proposed method performs poorly when applied to the ConvBig architecture on the CIFAR-10 dataset. Have the authors considered the reasons behind this and discussed them?

4. The quality of the figures can be further improved. For example, the text in Figures 3 and 4 is too small, and there are no detailed legends, which affects the readability and information transmission of the figures. Besides, the unit of Time should be provided in Table 2.

---

### Review · Reviewer_9bTx · 2024-07-25

**Summary Of Contributions:**

The authors use Convex Adversarial Polytope for Maxpool-based CNN to improve the verified bound.  This approach enables SOTA precision with a lower computational costs than other methods and therefore can be used on large-scale CNNs.

**Audience:**

Yes

**Broader Impact Concerns:**

No concerns

**Claims And Evidence:**

No

**Requested Changes:**

The introduction does a good job setting the stage as to the usecase within adversarial monitoring- I think it would be good to add a sentence within the abstract referencing this application to help those two sections flow together better.  It feels like it takes a long time to get to the main challenge which is handling maxpooling in a CNN for adversarial verification- right now each of those concepts are introduced separately.  I would try to bring them all together right up front, and then break it down as is done currently.  Right now it's clear from the abstract what the solution does, but not the "so what".  From the beginning of the intro, the "so what" is clear, but is a little disconnected from the abstract and maxpooling particular.  1.1 then gets into maxpooling.  Brining this all into 1-2 sentence in the abstract and start of the intro would be really beneficial.

Formatting- I know this might require fighting with the style file, but all of the results show up after the conclusion, which is a bit strange.  Perhaps after changes this will work itself out, but even force paginating might be appreciated to keep those closer to the text which references them.

Formatting for Tables 2/3 could be improved to include significant digits and units.  Perhaps put the variables as columns and the appraoches as rows? If space is an issue, convSmall could be condensed to "S" or "Sm" (and similar other changes) if needed.  The 100% accuracy could be dropped since it is the same for everything- just reference this in the text.
Tables 2/3: It feels like the accuracy(?) values should have more significant digits.
I would actually look to try and combine Tables 2/3 if possible because much of the information is duplicated.  Also the reader sees Table 2 first, where the proposed approach appears to lag SOTA, and only when they see Table 3 do they see the dramatic time difference.  Another possible formatting idea would be to move "convSmall"/Med/Large to a macro header and do sub-tables under that.  There are a variety of changes which could be made- I think it's worth spending some time how to draw the Reader's eyes to the most critical points so the take-aways are more visually obvious.

This is very small and only a personal suggestion (please disregard if you disagree).  I have trouble finding the reference to the equations in the text as "equations X" because the numbers blead into the other text.  Perhaps if formatted as "Eq. X", that capital-dense formatting migSht aid in visual readability.  Perhaps this is specified by the style file and can't be changed.

Minor- the x-axis values in Figures 3/4 are tough to read.  Perhaps change to scientific notation to make clearer.

**Strengths And Weaknesses:**

STRENGHTS:
* Overall the text/grammar is clear.  There are a few suggestions made below to provide additional high-level context, but the text is easy to read.
* The impact and importance of the work is made very clear.  The authors did a good job connecting a very focused algorithm around a very particular function to the broader ML community.
* The mathematical formalism is clear and easy to follow.  The Toy example in 2.2.1 is a nice addition.

WEAKNESSES:
* The computational performance of this approach is clearly evident in Table 3, but the performance does lag SOTA as stated in Table 2.  The abstract states that this approach allows SOTA precision, which is not supported by Table 3.  The result is still strong, but the abstract is overstated.  This language needs to be corrected- even changing to "near SOTA" results would be fine.
* Overall the results are very promising, but I believe could be made more thorough to be truly compelling.  A few specifics are given below, but as a summary, seeing more (real) datasets, complete results for those datasets, and multiple runs would be important.  Seeing other types of architecture (as opposed to just size variation) is also important.  The work just seems a bit incomplete as it is.
* It's fine to use simpler toy examples like MNIST and CIFAR10 to demonstrate the performance.  Given the claims that this enables quantification on larger CNNs, it would be good to demonstrate that on a larger (image size) dataset which would require larger models.
* Additionally showing the full plots (as in Figure 3) for MNIST in addition to CIFAR would be good.  Right now there are only detailed experiments on a single, toy, dataset.
* It would be helpful to run the experiments with multiple configurations to establish error bounds on the measurements.  It's not clear if a difference of 1% is significant (in Table 2), for example.
* The authors acknowledge the limitation that this study is conducted on only a specified architecture.  Even demonstrating on one other (not necessarily for every result, but a subset), would dramatically strengthen the result.
* Maxpooling is used in many architectures besides CNNs.  At a minimum it would be important to address this conceptually.  Ideally this could be explored in an experiment.
* Results formatting could be improved- see requested changes.
* Speaking to the slightly sub-SOTA performance seen in Table 3 is important.

---

### Review · Reviewer_8wrP · 2024-09-06

**Summary Of Contributions:**

The authors apply interval bounds propagation (IBP), essentially following (Wong & Kolter, 2018), to the verification of CNNs with max-pool layers.  This is achieved by "rolling-out" the max-pool as a series of transformations involving multiple ReLU operations, as attempted in prior studies.  Through evaluations on MNIST and CIFAR10, the authors show improvements on a DeepZ, DeepPoly, and PRIMA both in terms of the verification bounds and the running time.

**Audience:**

Yes

**Claims And Evidence:**

Yes

**Requested Changes:**

My understanding is that this is essentially an application of the IBP approach of Wong and Kolter (2018), where the contributions are primarily empirical, in terms of those evaluations on MNIST and CIFAR.

In that sense, it would be much more productive to focus the presentation on:
(1) Review  (Wong & Kolter, 2018)
(2) Adapt the approach to CNNs with max-pool layers
(3) Present the experimental evaluations
(4) Any remaining discussions or review of related works

Due to the status of the writing, I did not benefit from most of the presentation.  The takeaway for me is concentrated in the experimental results and their summaries.

I see there's a lot of content in the supplemental file, but there was no way for me to appreciate it as I read through the main paper.

**Strengths And Weaknesses:**

Strength:
=======
Successful demonstration of how the IBP approach of Wong and Kolter (2018) can be applied to CNNs with max-pool layers.

Weaknesses:
==========
The presentation deviates from standard submissions, where the introduction section attempts to fit the all of the related work review, the method, and the experiments, complete with forward references to supplemental material in a separate PDF file.  Reading through, the main method is presented through a toy example, simple enough, but does not adequately explain the dual formulation.  Key claims of contribution, singled out in the introduction, like the complexity bound follows immediately from (Wong & Kolter, 2018).

---

### Decision · Action_Editor_tqYR · 2024-12-01

**Recommendation:** Reject

**Comment:**

The reviewers identified several issues with empirical evaluation setup and authors agreed that the paper needs to be revised and resubmitted. I encourage authors to take the feedback of the reviewers into account.

**Audience:**

Yes

**Claims And Evidence:**

The reviewers identified several issues with empirical evaluation setup and authors agreed that the paper needs to be revised and resubmitted. I encourage authors to take the feedback of the reviewers into account.

**Resubmission Of Major Revision:**

The authors may consider submitting a major revision at a later time.